Identification of Flap endonuclease 1 as a potential core gene in hepatocellular carcinoma by integrated bioinformatics analysis

Li Chuanfei 1
Qin Feng 2
Hong Hao 3
Tang Hui 4
Jiang Xiaoling 5
Yang Shuangyan 4
Mei Zhechuan 1
Zhou Di zhoudi@cqmu.edu.cn 6
1 Department of Gastroenterology, The Second Affiliated Hospital of Chongqing Medical University , Chongqing , China
2 Department of Infectious Diseases, The People’s Hospital of Shi Zhu , Chongqing , China
3 Department of Orthopaedics, The Second Affiliated Hospital of Chongqing Medical University , Chongqing , China
4 Department of Infectious Diseases, Institute for Viral Hepatitis, The Key Laboratory of Molecular Biology for Infectious Diseases, Chinese Ministry of Education, The Second Affiliated Hospital of Chongqing Medical University , Chongqing , China
5 Tongnan District People’s Hospital, The First Affiliated Hospital of Chongqing Medical University , Chongqing , China
6 Department of Radiology, The First Affiliated Hospital of Chongqing Medical University , Chongqing , China
Uversky Vladimir
Electronic publication date: 2019 Sep 6
Publication date: 2019
Volume: 7
Electronic Location ID: e7619
Received 2019 Jun 13; Accepted 2019 Aug 5
Copyright: ©2019 Li et al.
Copyright year: 2019
Copyright holder: Li et al.
License: This is an open access article distributed under the terms of the Creative Commons Attribution License, which permits unrestricted use, distribution, reproduction and adaptation in any medium and for any purpose provided that it is properly attributed. For attribution, the original author(s), title, publication source (PeerJ) and either DOI or URL of the article must be cited.
License URL: https://creativecommons.org/licenses/by/4.0/

Keywords: Hepatocellular carcinoma, Core genes, Bioinformatics analysis, Flap endonuclease 1

Funding: National Natural Science Foundation of China 81300258 81801717 Medical Science Cultivation Fund of the First Affiliated Hospital of Chongqing Medical University PYJJ2018-15 This work was supported by the National Natural Science Foundation of China (No. 81300258, 81801717) and the Medical Science Cultivation Fund of the First Affiliated Hospital of Chongqing Medical University (PYJJ2018-15). The funders had no role in study design, data collection and analysis, decision to publish, or preparation of the manuscript.

==============================
Hepatocellular carcinoma (HCC) is a common yet deadly form of malignant cancer. However, the specific mechanisms involved in HCC diagnosis have not yet fully elucidated. Herein, we screened four publically available Gene Expression Omnibus (GEO) expression profiles (GSE14520, GSE29721, GSE45267 and GSE60502), and used them to identify 409 differentially expressed genes (DEGs), including 142 and 267 up- and down-regulated genes, respectively. The DAVID database was used to look for functionally enriched pathways among DEGs, and the STRING database and Cytoscape platform were used to generate a protein-protein interaction (PPI) network for these DEGs. The cytoHubba plug-in was utilized to detect 185 hub genes, and three key clustering modules were constructed with the MCODE plug-in. Gene functional enrichment analyses of these three key clustering modules were further performed, and nine core genes including BIRC5, DLGAP5, DTL, FEN1, KIAA0101, KIF4A, MCM2, MKI67, and RFC4, were identified in the most critical cluster. Subsequently, the hierarchical clustering and expression of core genes in TCGA liver cancer tissues were analyzed using the UCSC Cancer Genomics Browser, and whether elevated core gene expression was linked to a poor prognosis in HCC patients was assessed using the GEPIA database. The PPI of the nine core genes revealed an interaction between FEN1, MCM2, RFC4, and BIRC5. Furthermore, the expression of FEN1 was positively correlated with that of three other core genes in TCGA liver cancer tissues. FEN1 expression in HCC and other tumor types was assessed with the FIREBROWSE and ONCOMINE databases, and results were verified in HCC samples and hepatoma cells. FEN1 levels were also positively correlated with tumor size, distant metastasis and vascular invasion. In conclusion, we identified nine core genes associated with HCC development, offering novel insight into HCC progression. In particular, the aberrantly elevated FEN1 may represent a potential biomarker for HCC diagnosis and treatment.

Introduction

Hepatocellular carcinoma remains among the most common and deadly forms of cancer globally, posing a significant threat to human life (Forner, Reig & Bruix, 2018). HCC and other tumors develop as the result of the long-term accumulation of genetic mutations. Although a large number of biomarkers for the diagnosis of HCC have been identified (Turnbull, Sud & Houlston, 2018), the specific molecular mechanisms related to the onset, recurrence and treatment of HCC remain obscure. Therefore, it is essential to identify and exploit novel biomarkers involved in HCC onset and progression to better understand the pathogenesis of HCC.

Between human genome sequencing efforts and the rapid development of gene sequencing technologies, precision medicine has risen to prominence and been widely employed in the field of oncology (König et al., 2017; Nussinov et al., 2019). Precision medicine relies upon initially exploring potential therapeutic targets via high-throughput sequencing technologies (Robinson, 2012), as these technologies allow for the large-scale investigation of altered gene expression in the context of disease (Chen et al., 2010). However, sequencing results are often limited and inconsistent owing to the heterogeneity of samples in independent studies, and due to the fact that most studies focus on one cohort. As such, this study sought to analyze genes involved in liver cancer development using a range of available liver cancer-related gene chip datasets, with the goal of identifying potential novel molecular targets for liver cancer treatment and diagnosis.

For the purposes of this study, four HCC related Gene Expression Omnibus (GEO) database datasets were downloaded: GSE14520 (Roessler et al., 2012; Roessler et al., 2010), GSE29721 (Stefanska et al., 2011), GSE45267 (Chen et al., 2018c) and GSE60502 (Wang et al., 2014). By analyzing these four datasets, we identified 409 DEGs including 142 up-regulated and 267 down-regulated genes. The DAVID database was then used for a functional enrichment analysis of these DEGs, while the STRING database and Cytoscape were utilized to generate a protein-protein interaction (PPI) network, and three clustering modules were filtered out with the MCODE plug-in, among which clustering module 1 was most associated with HCC. In addition, nine core genes including BIRC5, DLGAP5, DTL, FEN1, KIAA0101, KIF4A, MCM2, MKI67, and RFC4, were identified within clustering module 1, and these corresponded to the hub genes in our PPI network. Using the GEPIA database, we performed survival analyses of patients based on expression of these nine core genes, revealing that their overexpression was linked to a poorer prognosis in HCC patients. When we surveyed the literature surrounding these genes, we found that all except for FEN1 had previously been confirmed to play a vital role in HCC. Existing studies have shown that FEN1 is highly expressed in various cancers, such as brain (Nikolova, Christmann & Kaina, 2009), lung (He et al., 2017; Zhang et al., 2018), breast (Abdel-Fatah et al., 2014), gastric (Wang, Xie & Chen, 2014), prostate (Lam et al., 2006) and pancreatic cancer (Isohookana et al., 2018), but its expression and role in HCC remains unclear. Our previous data indicated that FEN1 expression was elevated in HCC tumors, and this was confirmed upon comparing HCC samples and hepatoma cells to appropriate controls. Together, the results of these analyses suggest that FEN1 may be a core gene orchestrating the progression of HCC.

Materials & Methods

Data collection and DEG validation

Four liver cancer-related datasets (GSE14520, GSE29721, GSE45267 and GSE60502), were downloaded from GEO (http://www.ncbi.nlm.nih.gov/geo/). These datasets contained a total of 299 tumor samples and 289 non-tumor samples. GSE14520 contained 225 liver cancer samples and 220 adjacent controls; GSE29721 contained 10 pairs of liver cancer samples and adjacent control tissue; GSE45267 contained 46 liver cancer samples and 41 adjacent controls; and GSE60502 contained 18 pairs of liver cancer tissues and adjacent controls. The inclusive criteria for datasets were as follows: (1) the samples were from human HCC tissues and paired adjacent or non-tumor tissues; (2) gene expression profiling of mRNA; (3) each dataset included no less than ten paired samples; (4) the datasets were from 2009 to 2019. The exclusive criteria for datasets were as follows: (1) the HCC samples were induced by specific diseases, including chronic hepatitis B, hepatitis C, hepatitis D, alcoholic liver disease, nonalcoholic fatty liver disease and cholangiocarcinoma; (2) details of the samples were not available; (3) the datasets could not be analyzed by GEO2R; (4) no gene symbol. The specific platform information for these four datasets is compiled in Table 1. GEO2R was utilized to identify DEGs in these studies, using the screening criteria: |logFC(foldchange)| ≥ 1, P < 0.05, and adjusted P < 0.05. In addition, genes with multiple probe set or probe sets lacking matched gene symbols were removed or averaged, respectively. Then, the overall DEGs, as well as those that were up- and down-regulated in the four datasets were intersected and visualized using Funrich (v 3.0, http://funrich.org/index.html).

Table 1 Detailed information of the GEO datasets in this study.

Series accession	Species	Type	Platform	
GSE14520	Homo sapiens	Expression profiling by array	GPL3921 Affymetrix HT Human Genome U133A Array	
GSE29721	Homo sapiens	Expression profiling by array	GPL570 [HG-U133_Plus_2] Affymetrix Human Genome U133 Plus 2.0 Array	
GSE45267	Homo sapiens	Expression profiling by array	GPL570 [HG-U133_Plus_2] Affymetrix Human Genome U133 Plus 2.0 Array	
GSE60502	Homo sapiens	Expression profiling by array	GPL96 [HG-U133A] Affymetrix Human Genome U133A Array	
Notes.

GEO Gene Expression Omnibus

Functional enrichment analyses

Gene ontology analyses focuses on three domains: biological processes (BP), cellular components (CC), and molecular functions (MF), and such analyses are commonly used to understand the biological functions, pathways, or localization of DEGs. The Kyoto Encyclopedia of Genes and Genomes (KEGG) pathway analysis database surveys as a valuable resource for assessing how particular DEGs may be involved in or influenced by specific signaling pathways and disease states. The DAVID (https://david.ncifcrf.gov/) was utilized for functional enrichment analyses, with a P < 0.05 cutoff for significance.

PPI network analysis

The STRING database (v10.5; https://string-db.org/) was used to generate a DEG PPI, with a minimum interaction score cutoff of 0.4. Cytoscape (v 3.4.0, https://cytoscape.org/) was used for network visualization, and the CytoHubba plug-in was used to identify hub genes with the criteria of filtering degree ≥10. The MCODE plug-in was used to construct key clustering modules (MCODE score >10, degree cut-off = 2, node score cut-off = 0.2, Max depth = 100 and k-score = 2).

Validation of core genes

Hub genes among overall DEGs and the most critical clustering module were identified through the CytoHubba plug-in, and intersecting core genes were identified. The UCSC Cancer Genomics Browser (https://genome-cancer.ucsc.edu/) was then used for hierarchical clustering of these core genes. In addition, the expression profiles of these core genes in 421 TCGA liver cancer tissues, including 50 solid normal tissues and 371 primary tumors, were determined by analyzing available datasets. A core gene PPI network was constructed with the cBioportal online database (http://www.cbioportal.org/). Furthermore, correlations between the expression of FEN1 and MCM2, RFC4, or BIRC5 in TCGA liver cancer tissues were investigated. Finally, we used the ONCOMINE (https://www.oncomine.org/resource/login.html) and FIREBROWSE online database (http://firebrowse.org/) to investigate the FEN1 expression in various cancers including HCC.

Survival analysis

The GEPIA database (http://gepia.cancer-pku.cn/) was used to conduct survival analyses based on core gene expression, with hazard ratios (HRs) and 95% confidence intervals being calculated, and logrank P value <0.05 being the threshold of statistical significance.

Clinical samples

A total of 34 paired HCC tumor tissues as well as corresponding adjacent non-cancerous tissues were obtained from our Hospital’s Department of Hepatobiliary Surgery, with all patients providing informed consent. The content of the informed consent includes research purposes, risks and discomfort, benefits, and privacy issues. The form of the informed consent is in written. The study was examined and approved by the Second affiliated Hospital of Chongqing Medical University Ethics Committee (approval number: 2018-60).

Cell culture

Seven human liver cancer cell lines (SMMC-7721, BEL-7404, HCCLM3, HepG2, MHCC97-H, SK-HEP-1, and Huh-7) as well as normal human liver HL7702 cells were donated by the Institute for Viral Hepatitis, Chongqing Medical University. All cells were grown in high glucose DMEM (Gibco, Waltham, MA, USA) containing 10% FBS (Corning, Corning, NY, USA) at 37 °C in a 5% CO2 incubator.

Hematoxylin and eosin (H & E) staining

Paraffin-embedded tissues were dewaxed in xylene I, II, and III for 20 min each, and then dehydrated in an ethanol gradient (100%, 95%, 90%, 80%, and 70%), 3 min per step. Sections were then rinsed with distilled water for 5 min, and the nucleus was counterstained with hematoxylin for 3 min. Sections were washed again in water, followed by differentiation for 30 s in a 75% hydrochloric acid alcohol solution, and blue color was returned by washing with distilled water for 5 min. A red dye was then used for counterstaining for 5 min, after which samples were dehydrated in 70%, 80%, 95%, and 100% ethanol, 1 min per concentration. Sections were then cleared with xylene and sealed using neutral gum.

IHC analysis

IHC staining was conducted as previously described (Zhang et al., 2016). Briefly, paraffin-embedded sections were incubated at 56 °C for 2 h, and then 3% hydrogen peroxide was used for antigen retrieval. Afterwards, the sections were incubated with rabbit anti-human FEN1 (1:100, A1175, ABclonal, China) at four °C overnight. Then, sections were probed for 1 h using HRP-conjugated secondary antibodies at 37 °C, after which a DAB substrate kit was utilized, and hematoxylin was used for nuclear staining. The Image-Pro Plus (IPP) software (Media Cybernetics, Rockville, MD, USA) was used to quantify staining intensity. The frequency of positively-stained cells was determined on a 0–100 scale, while staining intensity was scored as follows: 0 = negative; 1 = weak; 2 = moderate; 3 = strong. These two scores were then multiplied together to yield an IHC score between 0 and 300. The final scores were assigned by two independent pathologists. The mean IHC score was used as a cutoff value to separate patients into low- and high-expression groups.

RT-qPCR analysis

TRIzol (ThermoFisher Scientific, Waltham, MA, USA) was used for total RNA extraction, and RNA was then reverse transcribed with the PrimeScript RT-PCR kit (Takara Bio, Dalian, China) based on provided protocols. The SYBR Premix Ex Taq II (Takara, Japan) kit was used to conduct RT-qPCR analysis on a Bio-Rad CFX96 Real-Time System (Bio-Rad, Hercules, CA, USA). The reaction conditions were as follows: pre-denaturation at 95 °C for 30 s, followed by 40 cycles of amplification at 95 °C for 5 s and 60 °C for 30 s, and a final cycle along the melting curve from 65 °C to 95 °C in increments of 0.5 °C for 5 s. mRNA levels were determined via the 2−ΔΔCt method, with GAPDH used for normalization. Primers used were: GAPDH F: 5′-GGTGGTCTCCTCTGACTTCAACA-3′ and R: 5′-GTTGCTGTAGCCAAATTCGTTGT-3′, FEN1 F: 5′-CTGTGGACCTCATCCAGAAGCA-3′ and R: 5′-CCAGCACCTCAGGTTCCAAGA-3′.

Statistical analysis

Statistical analyses and graphing were performed with SPSS v19.0 (SPSS Inc., Chicago, IL, USA) and Graph Pad Prism v8.0 (Graph Pad Software, San Diego, CA, USA), respectively. Data are means ± standard deviation (SD). Student’s t-tests were used to compare groups. Fisher’s exact test was used to assess correlations between the expression of FEN1 and HCC patient clinicopathological features. Spearman’s correlation analyses were used to compare the expression of pairs of genes in TCGA liver cancer tissues. P < 0.05 was the significance threshold (*P < 0.05, **P < 0.01).

Results

HCC-associated DEG identification

In this study, 1,088 total DEGs (505 and 583 up- and down-regulated, respectively) in GSE14520, 1,449 total DEGs (837 and 612 up- and down-regulated, respectively) in GSE29721, 1,604 total DEGs (713 and 891 up- and down-regulated, respectively) in GSE45267, and 1,533 total DEGs (792 and 741 up- and down-regulated, respectively) in GSE60502 were screened. Based on these datasets, a total of 409 overlapping DEGs were identified among these four datasets (142 and 267 up- and down-regulated, respectively), as visualized with the Funrich software (Fig. 1). DEGs are listed in Table S1.

Figure 1 Venn diagram.

(A) 409 DEGs were identified in four datasets (GSE14520, GSE29721, GSE45267, and GSE60502) via FUNRICH. These included 142 upregulated genes. (B) 267 downregulated genes. (C) Colors correspond to specific datasets, with intersecting areas indicating overlapping gene sets. DEG identification criteria were adj. P < 0.05 and |logFC(foldchange)| ≥ 1.

DEG functional enrichment analyses

To explore the biological activities of these DEGs, the DAVID database was used to conduct GO and KEGG enrichment analysis. With respect to BPs, up-regulated DEGs were primarily enriched in processes such as mitotic nuclear division, cell division, cell cycle, DNA replication, and mitotic sister chromatid segregation (Fig. S1A), while down-regulated DEGs were primarily enriched in processes such as redox process, the cytochrome 450 pathway, drug metabolism, and negative regulation of growth (Fig. S2A). With respect to CCs, up-regulated DEGs were primarily enriched in the nucleoplasm, nucleus, cytoplasm, spindle, and cellular intermediates (Fig. S1B), while down-regulated DEGs were mostly enriched in extracellular exosomes, organelle membranes, blood microparticles, extracellular regions, and the mitochondrial matrix (Fig. S2B). With respect to MFs, up-regulated DEGs were primarily enriched in functions such as protein binding, ATP binding, DNA helicase activity, protein kinase binding, single-stranded DNA binding, and chromatin binding (Fig. S1C), whereas down-regulated DEGs were primarily associated with iron ion binding, oxidoreductase activity, heme binding, monooxygenase activity, and oxygen binding (Fig. S2C).

A KEGG analysis revealed that up-regulated DEGs were particularly enriched in pathways such as the cell cycle, DNA replication, P53 signaling, and tumor pathways (Fig. S1D). However, down-regulated DEGs were mostly associated with metabolic pathways, fatty acid degradation, chemical carcinogenesis, and PPAR signaling (Fig. S2D).

PPI network and module analyses

To better understand interactions among DEGs, the STRING online database was used to generate a PPI network consisting of 403 nodes and 3,502 edges, which was visualized using Cytoscape. Six of the 409 DEGs were not included in this network (Fig. 2A). This network was then analyzed using the MCODE plug-in, and three clustering modules were filtered out according to the chosen screening conditions. Clustering module 1 scored 58.492 with 62 nodes and 1,784 edges (Fig. 2B), clustering module 2 scored 11.529 with 18 nodes and 98 edges (Fig. 2C), and clustering module 3 scored 10.917 with 25 nodes and 131 edges (Fig. 2D). The genes in clustering module 1 were up-regulated DEGs, whereas those in the other two modules were primarily down-regulated DEGs.

Figure 2 DEG PPI network and modular analysis.

(A) STRING was employed to create a PPI network of 403 nodes and 3,502 edges, visualized using Cytoscape software. Genes that are upregulated are shown by red nodes, while those that are downregulated are blue. The MCODE plug-in was used to analyze highlighted regions. (B) Clustering module 1 scored 58.492 with 62 nodes and 1,784 edges. (C) Clustering module 2 scored 11.529 with 18 nodes and 98 edges. (D) Clustering module 3 scored 10.917 with 25 nodes and 131 edges.

Functional enrichment analysis of key clustering modules

The DAVID database was next used to explore the biological functions of genes in these key clustering modules (Tables S2–S4). With respect to BPs, clustering module 1 was primarily enriched in cell differentiation, mitotic nuclear division, DNA replication, and DNA helicase activity, while clustering module 2 was primarily enriched in plasminogen activation, coagulation, cytolysis, and complement activation regulation, and clustering module 3 was primarily enriched in steroid metabolism, heterogeneous biomass metabolism, and exogenous drug catabolism. With respect to CCs, clustering module 1 was primarily enriched for the nucleoplasm, nucleus, intermediate, spindle, cytoplasm, nuclear chromosome, while clustering module 2 was primarily enriched for exosomes, extracellular regions, membrane attack complexes, and extracellular vesicles, and clustering module 3 was primarily enriched for organelle membranes, the endoplasmic reticulum membrane, and high-density lipoproteins. With respect to MFs, clustering module 1 was primarily enriched for protein binding, ATP binding, DNA helicase activity, protein kinase binding, and DNA binding, while clustering module 2 was primarily enriched for endopeptidase activity, transcription factor binding, steroid binding, RNA polymerase II transcription factor activity, and enzyme binding activity, and clustering module 3 was primarily enriched for aerobic binding, iron ion binding and heme binding.

A KEGG analysis revealed that clustering module 1 was primarily enriched in the cell cycle, oocyte meiosis, DNA replication, and p53 signaling, while clustering module 2 was primarily enriched in the complement system, prion disease, and systemic lupus erythematosus, and clustering module 3 was primarily enriched in chemical carcinogenesis, retinol metabolism, P450 drug metabolism, and metabolic pathways.

Identification of core genes and analysis of their clinical significance

Next, core genes involved in HCC were identified based on their levels of interaction via analyzing our PPI network using the Cytoscape program. Based on our clustering module analysis, clustering module 1 included 62 genes was found closely related to the progression of HCC. Then, nine total genes (BIRC5, DLGAP5, DTL, FEN1, KIAA0101, KIF4A, MCM2, MKI67, and RFC4) were identified based on the intersecting genes among the top 40 genes derived from 12 different algorithms by the cytoHubba plug-in (Table S5). In addition, 185 hub genes were identified across 403 nodes in our PPI network based on the filtering degree ≥10 criteria (Table S6). The nine identified core genes were found belonged to this larger subset of hub genes (Fig. 3A).

Figure 3 Identification of nine core HCC-associated genes.

(A) Nine core genes in clustering module 1 screened by 12 algorithms using the cytoHubba plug-in intersected with Hub genes. (B) Core gene expression profiles in 421 TCGA liver cancer tissues, including 50 solid normal tissues and 371 primary tumors, **P < 0.01. (C) The UCSC database was used for core gene hierarchical clustering.

To investigate core gene expression in HCC, a hierarchical clustering analysis was performed using the UCSC Cancer Genomics Browser, revealing that these nine core genes were highly expressed in most liver cancer samples (Fig. 3C). Next, the expression profiles of these nine core genes in 421 TCGA liver cancer tissues, including 50 solid normal tissues and 371 primary tumors, were downloaded and analyzed, revealing that the expression of these core genes was significantly elevated in HCC (Fig. 3B). The correlations between core gene expression levels and patient prognosis in 182 total HCC samples were further assessed, revealing that BIRC5 expression (HR = 2, logrank P = 6.7e−05) was correlated with worse overall survival (OS) for HCC patients, as was that of DLGAP5 (HR = 1.9, logrank P = 0.00039), DTL (HR = 1.7, logrank P = 0.0049), FEN1 (HR = 1.5, logrank P = 0.022), KIAA0101 (HR = 1.7, logrank P = 0.002), KIF4A (HR = 1.8, logrank P = 0.001), MCM2 (HR = 1.7, logrank P = 0.0022), MKI67 (HR = 1.9, logrank P = 0.00045), and RFC4 (HR = 1.7, logrank P = 0.004) (Fig. 4). Furthermore, the expression of BIRC5 (HR = 1.6, logrank P = 0.002) was correlated with decreased disease-free survival (DFS) for HCC patients, as was that of DLGAP5 (HR = 1.6, logrank P = 0.0033), DTL (HR = 1.6, logrank P = 0.0016), FEN1 (HR = 1.5, logrank P = 0.0075), KIAA0101 (HR = 1.6, logrank P = 0.0022), KIF4A (HR = 1.6, logrank P = 0.0011), MCM2 (HR = 1.6, logrank P = 0.0034), MKI67 (HR = 1.9, logrank P = 4.2e−05), and RFC4 (HR = 1.5, logrank P = 0.011) (Fig. 5). High expression of these nine core genes was associated with significantly reduced survival among HCC patients.

Figure 4 Association between core genes and overall survival (A, BIRC5; B, DLGAP5; C, DTL; D, FEN1; E, KIAA0101; F, KIF4A; G, MCM2; H, MKI67 and I, RFC4) in those with HCC.

CI, confidence interval; HR, hazard ratio. High- and low-risk groups are shown in red and blue, respectively. P < 0.05 was the significance threshold.

Figure 5 Association between core genes and disease-free survival (A, BIRC5; B, DLGAP5; C, DTL; D, FEN1; E, KIAA0101; F, KIF4A; G, MCM2; H, MKI67 and I, RFC4) in those with HCC.

CI, confidence interval; HR, hazard ratio. High- and low-risk groups are shown in red and blue, respectively. P < 0.05 was the significance threshold.

FEN1 may be a key candidate gene in HCC

To clarify the PPI network for these nine core genes, the cBioportal online database were explored and identified an interaction between FEN1, MCM2, BIRC5 and RFC4 (Fig. 6A). Previous studies have confirmed that MCM2, BIRC5, and RFC4 are abnormally highly expressed in HCC, and that they participate in the regulation of HCC tumor biology. However, the expression of FEN1 and its clinical significance in HCC is unclear. Therefore, the correlations between the expression of FEN1 and these three other genes were further analyzed. The expression of FEN1 in TCGA liver cancer tissues was positively correlated with that of MCM2 (r = 0.853, P = 0.000), BIRC5 (r = 0.809, P = 0.000), and RFC4 (r = 0.852, P = 0.000) (Figs. 6B–6D), suggesting that FEN1 may play as important a role in the progression of liver cancer as do MCM2, BIRC5, and RFC4. The ONCOMINE and FIREBROWSE databases were further utilized to investigate the expression of FEN1 in various cancers, including HCC. The results revealed that FEN1 expression was clearly elevated in most cancers, including bladder, breast, colorectal, esophageal, lung, and liver cancer (Figs. 7A and 7B). In addition, FEN1 was found overexpressed in three HCC-related datasets (Fig. 7C).

Figure 6 Correlations between the expression of FEN1 and MCM2, BIRC5, and RFC4.

(A) A PPI network for the nine core genes generated with the Cbioportal database. (B) The correlation between FEN1 and MCM2 in 421 TCGA liver cancer tissues including 50 solid normal tissues and 371 primary tumors. (C) The correlation between FEN1 and BIRC5 in 421 TCGA liver cancer tissues including 50 solid normal tissues and 371 primary tumors. (D) The correlation between FEN1 and RFC4 in 421 TCGA liver cancer tissues including 50 solid normal tissues and 371 primary tumors. P < 0.05 was the significance threshold.

Figure 7 The expression of FEN1 in cancers including HCC.

The expression of FEN1 in various tumor tissue types was analyzed in the ONCOMINE (A) and FIREBROWSE databases (B). (C) The expression of FEN1 in three different HCC-related chip data sets was analyzed in the ONCOMINE database.

Experimental validation

In this study, 34 paired HCC and adjacent control tissues were used to verify the expression of FEN1 in HCC. The pathologic diagnosis of HCC and matched adjacent samples were confirmed by H&E staining (Figs. 8A and 8C). FEN1 staining was localized to the nucleus and cytoplasm, with stronger expression in the HCC samples relative to the adjacent controls (Figs. 8B and 8D). Furthermore, IHC analysis indicated that FEN1 levels were significantly higher in HCC relative to adjacent tissues (Fig. 8E), with IHC scores being significantly higher in HCC samples (Fig. 8F).

Figure 8 FEN1 was up-regulated in HCC tissues and hepatoma cell lines.

H & E staining of adjacent tissue (A) and HCC tissue (C). (B) FEN1 IHC in adjacent tissue. (D) FEN1 IHC in HCC tissue. (E) FEN1 IHC staining quantification; n = 34/group. (F) FEN1 IHC scores; n = 34/group. (G) FEN1 expression in control HL7702 liver cells and in hepatoma cell lines (as indicated); *P < 0.05, **P < 0.01.

The correlation between FEN1 expression and HCC patient clinicopathological features was investigated via Fisher’s exact test. As shown in Table 2, there were significant correlations between FEN1 expression and tumor size (P = 0.047 < 0.05), metastasis (P = 0.013 < 0.05) and vascular invasion (P = 0.024 < 0.05). FEN1 expression did not significantly correlated with gender, age, tumor multiplicity, TNM stage, pathological grade, HBsAg, liver cirrhosis or serum alpha-fetoprotein (AFP) (P > 0.05). Furthermore, FEN1 mRNA levels were found significantly elevated in six human hepatoma cell lines relative to that in the normal human liver cell line, HL7702 (Fig. 8G).

Table 2 Correlation between FEN1 expression and clinicopathological features in 34 paired HCC patients.

Clinicopathological features	Cases (n = 34)	FEN1 expression	P value	
		High (%)	Low (%)		
All case (n = 34)	34	25	9		
Gender				1.000	
Male	29	21 (61.8%)	8 (23.5%)		
Female	5	4 (11.8%)	1 (2.9%)		
Age (y)a				0.697	
<52	15	12 (35.3%)	3 (8.8%)		
≥52	19	13 (38.2%)	6 (17.6%)		
Tumor size				0.047*	
<5 cm	19	11 (32.4%)	8 (23.5%)		
≥5 cm	15	14 (41.2%)	1 (2.9%)		
Tumor multiplicity				0.348	
Single	27	21 (61.8%)	6 (17.6%)		
Multiple	7	4 (11.8%)	3 (8.8%)		
TNM stage				1.000	
I ∼ II	23	17 (50.0%)	6 (17.6%)		
III ∼ IV	11	8 (23.5%)	3 (8.8%)		
Pathological grade				1.000	
Well/moderate	27	20 (58.8%)	7 (20.6%)		
Poor	7	5 (14.7%)	2 (5.9%)		
Metastasis				0.013*	
With	12	12 (35.3%)	0 (0.0%)		
Without	22	13 (38.2%)	9 (26.5%)		
HBsAg				1.000	
Positive	30	22 (64.7%)	8 (23.5%)		
Negative	4	3 (8.8%)	1 (2.9%)		
Liver cirrhosis				0.687	
With	12	8 (23.5%)	4 (11.8%)		
Without	22	17 (50.0%)	5 (14.7%)		
Serum AFP				0.725	
>400 ng/ml	18	12 (35.3%)	6 (17.6%)		
⩽400 ng/ml	16	9 (26.5%)	7 (20.6%)		
Vascular invasion				0.024*	
Present	14	13 (38.2%)	1 (2.9%)		
Absent	20	11 (32.4%)	9 (26.5%)		
Notes.

a Patients were divided according to the median age.

AFP alpha-fetoprotein

HBsAg hepatitis B surface antigen

* P < 0.05.

Discussion

The occurrence and progression of HCC are complex, with multiple cumulative genetic changes ultimately culminating in progressive disease. High-throughput technologies such as gene chips have been widely employed to elucidate the underlying mechanisms, providing an innovative and effective approach to the diagnosis, prevention, and treatment of HCC. Numerous studies have been conducted to clarify genetic changes underlying HCC development, but results to date remain inconclusive of incomplete. As such, there is further need to investigate the molecular mechanisms governing HCC.

Functional analysis of DEGs and screening of core genes: in this study, we identified 409 total DEGs (142 and 267 up- and down-regulated, respectively) shared among four HCC datasets, and these genes were used for functional enrichment analyses. A GO analysis revealed the up-regulated DEGs to be primarily linked with cell division, the cell cycle, and DNA replication, whereas down-regulated DEGs were mostly associated with redox reactions, cytochrome 450 functionality, and negative growth regulation. A KEGG pathway analysis revealed up-regulated DEGs to mostly be associated with signaling relating to the cell cycle, DNA replication, p53 signaling, and tumor pathways, whereas down-regulated DEGs were mostly linked to metabolic pathways such as fatty acid degradation. These results suggested that the up-regulated DEGs may affect HCC progression via regulating DNA replication and the cell cycle, whereas down-regulated DEGs may be linked to HCC progression through metabolic pathways. Previous studies have indicated that the dysregulation of the cell cycle is a key hallmark of many cancer types (Hydbring et al., 2017). As one of the most important tumor suppressor genes, p53 is closely related to tumorigenesis, with at least 50% of cancer patients exhibiting p53 mutations or loss of function (Kandoth et al., 2013; D’Orazi & Cirone, 2019; Kim, Zhang & Lozano, 2015). P53 signaling dysregulation has repeatedly been confirmed to be linked with cancer development (Levrero & Zucman-Rossi, 2016). There is also increasing evidence that metabolism regulates cancer growth and proliferation (Zhu & Thompson, 2019; Saito et al., 2016). We further used our DEGs to generate a PPI, which consisted of three key clustering modules screened using the MCODE plug-in. Functional enrichment analyses revealed that clustering module 1 was closely related to gene mutations in the progression of HCC, and was enriched in genes linked to the cell cycle, DNA replication, and the p53 signaling pathway. Then nine core genes (BIRC5, DLGAP5, DTL, FEN1, KIAA0101, KIF4A, MCM2, MKI67 and RFC4) in clustering module 1 were screened, as these genes were also hub genes in the overall PPI network. There are numerous studies related to HCC chips that have sought to identify the molecules involved in the onset and progression of HCC. However, due to the different data sets and search strategies included in each study, the key molecules identified are different. Nevertheless, the core genes we identified were similar to those found by other researchers. Cai, Wang & Tu (2019) identified twelve hub genes including BIRC5 and MCM2 in HCC. DTL and MCM2 were also identified as hub genes for HCC in another study (Sang et al., 2018). Shen et al. (2019) identified FEN1 and RFC4 as core genes by analyzing four HCC datasets. The above studies indicated that the core genes screened and identified in our study were of value. Afterwards, the hierarchical clustering and expression profiles of core genes in TCGA liver cancer tissues were analyzed using the UCSC Cancer Genomics Browser, and survival analyses suggested that aberrantly high expression of these core genes was predictive of a poor HCC patient prognosis, suggesting these core genes may be key molecular biomarkers for HCC diagnosis and treatment.

Research progress of core genes in HCC: BIRC5, also known as survivin, is an anti-apoptotic protein reported to function as a potential oncogene in the context of many cancers (Duffy et al., 2007). Studies have shown that BIRC5 is highly expressed in the vast majority of tumors, including HCC (Su, 2016). In addition, elevated BIRC5 levels have been found to be associated with histological grade, tumor size, and TNM stage in HCC patients (Chen et al., 2019). DLGAP5, also known as HURP, is a cell cycle regulatory gene that has been found to be highly expressed in liver cancer (Chang, Lin & Yeh, 2011; Tsou et al., 2003). Liao et al. (2013) reported the abnormally high expression of DLGAP5 in HCC, and found that this was related to promoter methylation level, with the silencing of DLGAP5 significantly inhibiting cell proliferation, migration, and colony formation in vitro. DLGAP5 knockdown inhibited the proliferation of hepatoma cells by reducing P53 accumulation (Kuo et al., 2012). DTL is a substrate receptor for the CRL4 ubiquitin ligase, serving as a key regulator of the cell cycle and genomic stability. DTL upregulation in invasive HCC has been found to be positively correlated with tumor grade and patient survival (Chen et al., 2018b; Pan et al., 2006). FEN1, a structurally specific metal nuclease, plays a vital role in DNA damage repair (Lieber, 1997) and maintenance of genomic stability (Becker et al., 2018). FEN1 has been found to be expressed at high levels in several cancer types, including those of the lung (He et al., 2017), breast (Abdel-Fatah et al., 2014), gastric (Wang, Xie & Chen, 2014), prostate (Lam et al., 2006) and pancreatic (Isohookana et al., 2018). FEN1 mutations have been associated with the occurrence of gastrointestinal tumors, including HCC (Liu et al., 2012b), suggesting that FEN1 may be important in the development of gastrointestinal tumors. Nevertheless, the expression and biological function of FEN1 in HCC remains unclear. KIAA0101 is a nuclear antigen-associated factor that is present in proliferating cells and which regulates proliferative processes. KIAA0101 has been found to be highly expressed in HCC, representing a potential biomarker for this tumor type that is relevant to disease treatment and patient prognosis (Yuan et al., 2007). KIAA0101 variant 1 can promote the survival of liver cancer cells by regulating p53 functionality, indicating that inhibition of KIAA0101 variant 1 may be a promising therapeutic strategy (Liu et al., 2012a). Numerous studies have revealed that KIF4A is an oncogene in the context of lung, oral, and breast cancer. High KIF4A expression was significantly correlated with tumor stage, tumor differentiation, and metastasis, and may be a biomarker for poor HCC prognosis (Hou et al., 2017). In addition, KIF4A promoted the proliferation and invasion of hepatoma cells via p53 and Akt signaling (Hou et al., 2017; Huang et al., 2018). MCM2 has been shown to be critical for chromatin composition and for the focal formation of p53 binding protein 1 in HepG2 cells (Chen et al., 2018a). High MCM2 expression has been associated with poor prognosis in HCC patients (Liu et al., 2018). MKI67 encodes a nuclear antigen expressed during the G1, S and G2-M phases in proliferating cells, and its expression levels are closely related to tumor growth rate (Maeda et al., 1995), histological stage (Ng et al., 1995), and tumor recurrence for HCC (Shirabe et al., 1995). Higher MKI67 expression was associated with both faster HCC progression and a poorer patient prognosis (Luo et al., 2015). Consistent with our results, RFC4 has been found to be overexpressed in HCC (Arai et al., 2009). The silencing of RFC4 expression was able to reduce HepG2 cell proliferation and promote apoptosis, and was also associated with the increased sensitivity of cells to doxorubicin and camptothecin, suggesting RFC4 may be a novel target for liver cancer treatment.

The identification of FEN1 and its potential mechanisms in HCC: as these past results show, all of these core genes, except for FEN1, have been reported to play a role in HCC progression. To further explore interactions among these nine core genes, a PPI network was generated revealing interactions between FEN1, MCM2, RFC4, and BIRC5. As expected, FEN1 expression was significantly positively correlated with that of MCM2, RFC4, and BIRC5 in TCGA liver cancer tissues, suggesting that FEN1 may be as involved in the development of HCC as are MCM2, RFC4, and BIRC5. Previous studies had confirmed that FEN1 was aberrantly up-regulated in various cancers. However, the expression of FEN1 in HCC remain obscure. Therefore, we first utilized the ONCOMINE and FIREBROWSE online databases to investigate the expression of FEN1 in HCC, and found that FEN1 was highly expressed in multiple tumor tissues including those of HCC patients, consistent with previous reports. We then verified this in HCC tissues from 34 patients via IHC staining. Interestingly, a correlation analysis revealed that high expression of FEN1 was significantly correlated with tumor size, metastasis and vascular invasion, suggesting that FEN1 may be involved in the regulation of liver cancer proliferation and metastasis. Similarly, a study had confirmed that the expression of FEN1 in gastric cancer was positively associated with the degree of differentiation, lymphatic metastasis, tumor size and TNM stage (Wang, Xie & Chen, 2014). In addition, the FEN1 was up-regulated in lung cancer cells and associated with the clinical lung cancer stage (He et al., 2017). Another study also suggested that FEN1 might contribute to nodal metastasis in lung adenocarcinoma and result in a poor prognosis (Hwang et al., 2015). These findings suggested that FEN1 may be involved in the regulation of multiple tumor biological processes. Mechanistically, FEN1, a structure-specific metallonuclease, is involved in DNA replication, DNA synthesis, DNA damage repair, non-homologous end joining and homologous recombination (Lieber, 1997; Wu, Wilson & Lieber, 1999). Importantly, FEN1 is essential for maintaining genome stability (Becker et al., 2018). The upstream and downstream molecules of FEN1 may be involved in the regulation of various tumors. A study has previously reviewed three underlying mechanisms by which FEN1 may be involved in tumorigenesis (Zheng et al., 2011). First, the interactions between FEN1 and other DNA metabolic proteins are formed. Second, the super-accumulated FEN1 in nucleolus and mitochondrion is thought to play a vital role in maintaining the stability of tandem ribosomal DNA repeats and in mitochondrial DNA replication. Third, FEN1 could be post-translationally modified through acetylation, methylation, or phosphorylation, and this was important for its ability to regulate nuclease activities and protein partner selection. One study reported that FEN1 knockdown was able to induce DNA damage and enhance the activation of p53 in lung cancer cells (He et al., 2017). Furthermore, FEN1 was found be regulated by the transcriptional repressor Ying Yang 1 (YY1) in response to DNA damaging agents in the context of breast cancer. Another study demonstrated that FEN1 promoted breast cancer cell proliferation via an epigenetic mechanism whereby FEN1-mediated up-regulation of DNMT1 and DNMT3a (Zeng et al., 2019). At present, whether FEN1 participates in HCC progression and what the underlying mechanisms of such participation may be remains unclear. We speculate that it may be related to the above mechanisms. In addition, our results showed that FEN1 was elevated in six human hepatoma cell lines relative to a control cell line via RT-qPCR analysis, which provided some basis for further study of the function of FEN1 in liver cancer. However, the specific biological function of FEN1 in HCC remains to be further investigated.

Conclusions

In summary, we screened 409 HCC-associated DEGs, of which 142 and 267 were up- and down-regulated, respectively. We identified nine core genes, including BIRC5, DLGAP5, DTL, FEN1, KIAA0101, KIF4A, MCM2, MKI67, and RFC4, that were up-regulated in HCC and that may play important roles in the development or progression of this cancer. We further confirmed the expression of FEN1 in HCC, suggesting that FEN1 may have potential as a new biomarker for HCC diagnosis, treatment, or prognosis determination. Our results offer significant improvements to current understanding of HCC pathogenesis. Additional studies will be needed to validate our findings, and to confirm whether and how FEN1 regulates HCC.

Supplemental Information

Figure S1 Results of GO and KEGG analyses of upregulated DEGs

Enriched biological processes (A), cellular components (B), and molecular functions (C). (D) Enriched KEGG pathways among upregulated DEGs. GO terms/KEGG pathways are shown on the x-axis, while y-axes detail x –log10 (P values) and numbers of DEGs.

Click here for additional data file.

Figure S2 Results of GO and KEGG analyses of downregulated DEGs

Enriched biological processes (A), cellular components (B), and molecular functions (C). (D) Enriched KEGG pathways among downregulated DEGs. GO terms/KEGG pathways are shown on the x-axis, while y-axes detail x –log10 (P values) and numbers of DEGs.

Click here for additional data file.

Table S1 A total of 409 DEGs were identified from four profile datasets, including 142 upregulated and 267 downregulated genes in HCC compared with normal tissues

Click here for additional data file.

Table S2 Gene function analysis of clustering module 1

Click here for additional data file.

Table S3 Gene function analysis of clustering module 2

Click here for additional data file.

Table S4 Gene function analysis of clustering module 3

Click here for additional data file.

Table S5 The core genes in clustering module 1were screened by 12 algorithms of cytoHubba plug-in

Click here for additional data file.

Table S6 185 Hub genes were screened and identified by CytoHubba plugin

Click here for additional data file.

Data S1 Raw data: The IHC intensity of FEN1 in HCC

Click here for additional data file.

Data S2 Raw data: The IHC score of FEN1 in HCC

Click here for additional data file.

Data S3 Raw data: Relative FEN1 mRNA expression in cell lines

Click here for additional data file.

Additional Information and Declarations

Competing Interests

Author Contributions

Human Ethics

Microarray Data Deposition

Data Availability

The authors declare there are no competing interests.

Chuanfei Li performed the experiments, analyzed the data, authored or reviewed drafts of the paper, approved the final draft.

Feng Qin analyzed the data, approved the final draft.

Hao Hong contributed reagents/materials/analysis tools, approved the final draft.

Hui Tang performed the experiments, contributed reagents/materials/analysis tools, prepared figures and/or tables, approved the final draft.

Xiaoling Jiang prepared figures and/or tables, approved the final draft.

Shuangyan Yang performed the experiments, prepared figures and/or tables, approved the final draft.

Zhechuan Mei authored or reviewed drafts of the paper, approved the final draft.

Di Zhou conceived and designed the experiments, approved the final draft.

The following information was supplied relating to ethical approvals (i.e., approving body and any reference numbers):

The Second affiliated Hospital of Chongqing Medical University Ethics Committee granted approval to carry out the study within its facilities (Ethical Application Ref: 2018-60).

The following information was supplied regarding the deposition of microarray data:

Data is available at the Gene Expression Omnibus database: GSE14520, GSE29721, GSE45267, GSE60502.

The following information was supplied regarding data availability:

The raw measurements are available in Supplemental Files.

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
