# Peer review of "Identification of Flap endonuclease 1 as a potential core gene in hepatocellular carcinoma by integrated bioinformatics analysis"

_PeerJ, doi:10.7717/peerj.7619_

## Round 0.1 · original submission · Major Revisions

All critiques of both reviewers should be addressed and the manuscript should be revised accordingly.

Reviewer 1 ·

Basic reporting

Clear and unambiguous, professional English used throughout.

Experimental design

Research question well defined, relevant & meaningful. It is stated how research fills an identified knowledge gap

Validity of the findings

Conclusions are well stated, linked to original research question & limited to supporting results

Additional comments

This manuscript “Identification of Flap endonuclease 1 as a potential core gene in hepatocellular carcinoma by integrated bioinformatics analysis” by Li et al tried to investigate the specific mechanisms involved in HCC diagnosis by integrated bioinformatics.
In this manuscript, the authors identified Nine core genes in the most critical cluster, and the PPI of the nine core genes revealed an interaction between FEN1, MCM2, RFC4, and BIRC5. Furthermore, FEN1 expression was positively correlated with that of three other core genes in TCGA liver cancer tissues. The goal of the study is clear and well stated. Background studies about the article is clearly articulated. Yet, there are still some problems as follows. I would like to suggest a major revision for it.
The detail comments as followings:

1. Line 35, should add one blank space before 185.
2. Line 43, the meaning of this sentence is a little confused, please make it clear.
3. Line 59, this sentence is confusing, please revise it.
4. Lines 74-75, the description of this sentence is not accurate, please correct it.
5. Line 156, change CO2 to CO2.
6. Line 172, delete the blank space after 56, including others in the whole manuscript.
7. Lines 184-192, should provide the detail parameter for the RT-PCR analysis.
8. Line 200, should add one blanks space before and after <.
9. Line 204, author should use third person mood to describe the whole manuscript, please correct it, including others in the whole manuscript.
10. Line 328, this section, authors should add several subtitles for discussion focus on the main issues.
11. Line 433, author should unify the references style, including the journal name and others. Such as, line 487 and line 492 ect.

Reviewer 2 ·

Basic reporting

No comment.

Experimental design

1. There are lots of datasets related to HCC in GEO database, please explain why the four GSE datasets were selected, especiall datasets contained small number of samples, such as GSE29721 and GSE60502. Maybe the authors should list the inclusive and exclusive criteria.
2. In the exploration of correlation between FEN1 and clinicopathological features, please add some other crucial clinicopathological factors, such as MVI, AFP level, etc. These traits are also close related to HCC prognosis.

Validity of the findings

No comment.

Additional comments

Using bioinformatic analysis, the authors have obtained nine core genes related to HCC progression and prognosis, in addition, they made a further exploration in the effect of FEN1 dysregulation on HCC. The study was well written. Besides the comments in the 3 areas above, I have several comments:
1. Most of the nine core genes have been reported in previous studies, maybe the authors could make a comparison with the previous bioinformatic studies.
2. In the "Discussion" part, the authors should make further discussion about the potential mechanisms of correlation between FEN1 and clinicopathological features. In addition, the discussion about the potential mechanisms of FEN1 in HCC progression and prognosis is not well discussed.

---

## Round 0.2 · accepted · Accept

Since critiques of the reviewers were adequately addressed and the manuscript was revised accordingly, the amended version is acceptable now.

Reviewer 2 ·

Basic reporting

None.

Experimental design

None.

Validity of the findings

None.

Additional comments

Thanks for the authors' response. I have learned a lot from them, and I don not have any questions for this manuscript.